# Foam Quality of Foams Formed on Capillaries and Porous Media Systems

**Victor Starov** [1,*] , **Anna Trybala** [1] , **Phillip Johnson** [1] **and Mauro Vaccaro** [2]

1   Department of Chemical Engineering, Loughborough University, Loughborough LE11 3TU, UK;
A.Trybala@lboro.ac.uk (A.T.); P.Johnson@lboro.ac.uk (P.J.)
2   Procter & Gamble, Temselaan 55, Grimbergen, B-1853 Brussels, Belgium; vaccaro.m@pg.com
*   Correspondence: V.M.Starov@lboro.ac.uk

**Abstract:** Foams are of great importance as a result of their expansive presence in everyday life—they are used in the food, cosmetic, and process industries, and in detergency, oil recovery, and firefighting. There is a little understanding of foam formation using soft porous media in terms of the quality of foam and foam formation. Interaction of foams with porous media has recently been investigated in a study by Arjmandi-Tash et al., where three different regimes of foam drainage in contact with porous media were observed. In this study, the amount of foam generated using porous media with surfactant solutions is investigated. The aim is to understand the quality of foam produced using porous media. The effect of capillary sizes and arrangement of porous in porous media has on the quality of foam is investigated. This is then followed by the use of soft porous media for foam formation to understand how the foam is generated on the surface of the porous media and the effect that different conditions (such as concentration) have on the quality of the foam. The quality of foam is a blanket term for bubble size, liquid volume fraction, and stability of the foam. The liquid volume fraction is calculated using a homemade dynamic foam analyser, which is used to obtain the distribution of liquid volume fraction along with the foam height. Soft porous media does not influence substantially the rate of decay of foam produced, however, it decreases the average diameter of the bubbles, whilst increasing the range of bubble sizes due to the wide range of pore sizes present in the soft porous media. The foam analyser showed the expected behaviour that, as the foam decays and becomes drier, the liquid volume fraction of the foam falls, and therefore the conductivity of foam also decreases, indicating the usefulness of the home-made device for future investigations.

**Keywords:** foams; porous media; capillaries; foam quality; SDS; Tween-20



## 1. Introduction

Foams are multiphase colloidal systems that are generated by air entrapment within the solution because with pure water, these bubbles are unstable and rapidly coalesce [1–6]. This means that a stabilising agent, such as surfactants, must be added to create stable foams that are required for a wide range of applications including food, pharmaceutical, firefighting, enhanced oil recovery, mining, soil remediation, detergents, and cosmetic industries [5,7–12]. Foams interact with porous media and are produced by porous media in many applications [13–22]. As discussed in [13,19], foam interaction and production is of great interest in the household cleaning industry in particular dishwashing, although this work can also apply to multiple cleaning products including personal cleaning products due to the porous nature of the human skin [11]. Another industry which foam interaction and formation with porous media is enhanced oil recovery (EOR) which has been investigated in [15], where foam quality is investigated to see its effect on sweep efficiency within the porous media and how foam, in general, can improve the sweep efficiency for the extraction of oil from the porous rocks. Further work on the effect of foam on the extraction

from porous rock has been conducted in [16,17]; both studies also concentrated on understanding what properties affect foam quality and achieved the optimum sweep through the media. How foam interacts with porous media has only recently been investigated in more detail and there are many questions that remain for this system [13,23–27]. The first model of foam drainage of foam in contact with porous media was introduced in a study by Arjmandi-Tash et al. [19,28,29], where the authors showed that there are three different regimes of drainage process. The first regime is rapid imbibition in which the drainage due to gravity is slower than the drainage caused by capillary imbibition into the porous media, meaning that there is no opportunity for a liquid level to form on top of the porous media and that the bottom of the foam never reaches a critical liquid volume fraction value. The next regime is known as the intermediate imbibition; this is when the rate of drainage caused by gravity is comparable with the rate of imbibition leading to the bottom of the foam to reach maximum liquid volume fraction below critical, but there is no liquid layer formed on top of the porous media. The final case is slow imbibition in which the rate of drainage caused by gravity is greater than that of the imbibition into the porous media leading to the creation of a liquid layer on the top of the porous media. This theory was extended and verified in [14], where foam was deposited on thin porous substrates were the three regimes discussed were observed.

In addition to the interaction of porous media with foams, recent studies on how foams are formed with the help of porous media have also been conducted [13,23,24]. In these investigations, the amount of foam mass-produced with commercial dishwashing solution and sodium dodecyl sulphate (SDS) solutions was absorbed into soft porous media. In these experiments, it was found that for commercial products, the amount of foam produced reaches a maximum of 60–80% commercial product to 20–40% water, indicating that the maximum amount of foam is produced with a low amount of liquid [23]. For SDS solutions, it was discovered that the amount of foam produced is maximum after 10 times the critical micelle concentration, indicating that micelles are an important property in terms of foam production [13]. These experimental investigations led to the compression/decompression system model which reliably predicts the amount of foam produced per compression of a soft porous media for both commercial products and SDS solutions [24].

The property that requires further investigation is the quality of foam, which is a blanket term for the bubble diameter, liquid volume fraction, and stability of the foam. For different applications these properties have different preferred values; for example, for drug delivery, a foam applied to the skin requires to be stable for a prolonged period of time so that the liquid does not drain too quickly and have a high enough liquid volume fraction so that the right amount of the drug is applied. In contrast, for firefighting, foam requires a lot lower stability so that the liquid can rapidly put out the fire. Foam quality for a commercial product requires the bubble diameter to be homogenous so that is appealing for the consumer, a large liquid volume fraction so that a small amount of product has to be used to achieve the required results, and has to be stable enough to be applied but not too stable as to be difficult to remove after cleaning.

Below, the quality of foam of surfactant solutions is investigated for multiple systems to help understand the best way to discuss the foam quality. The components of foam quality are discussed and investigated for each of the systems, the work shown here helps build a model for foam quality similar to what was achieved for the amount of foam previously [23,24].

## 2. Materials and Methods

### 2.1. Foam Column

Investigations of the quality of foam produced by air injection were undertaken using a foam column manufactured in-house for the project, as shown in Figure 1. This rig consists of an acrylic column attached to a removable base in which the different substrates are interchanged.

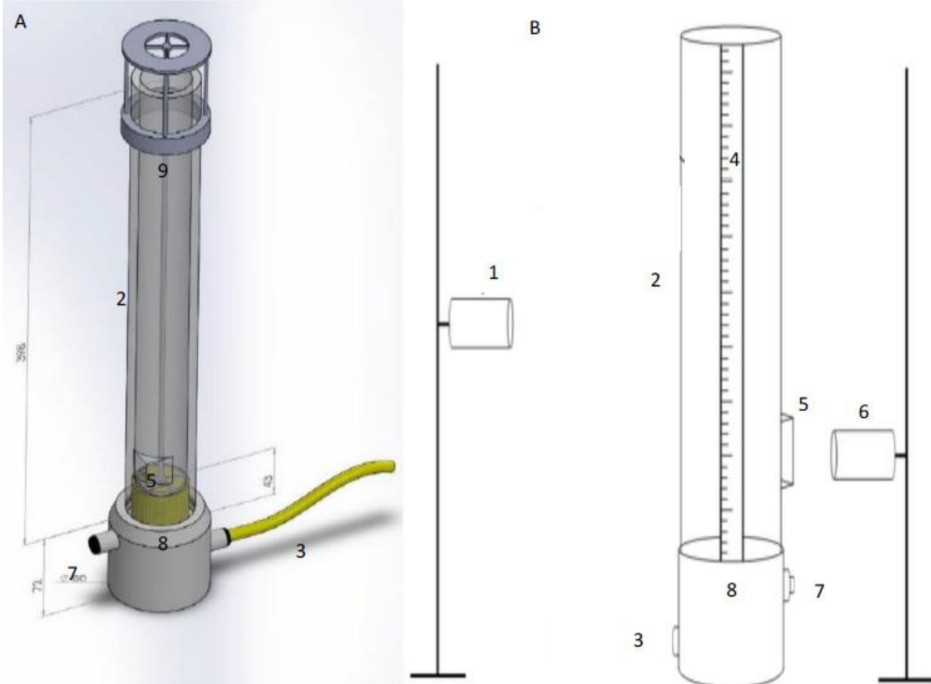

**Figure 1.** Drawing of the foam column used in the experiments. (**A**) shows a depiction of how the column will look when porous media is used instead of capillaries; a metal holder has to be present to prevent unwanted ejection of the media from the column, due to pressure build-up beneath the media. (**B**) shows a schematic of the experimental setup.

In Figure 1, the section labels are as follows: 1. the camera that is used to measure foam height, 2. the acrylic column, 3. gas inlet, 4. scale, 5. prism used to provide a flat interface to measure bubble diameter which is unaffected by the curvature of the column, 6. the camera used to measure the bubble diameter, 7. liquid outlet which is plugged during the experiment and is used to allow the easy drainage of the liquid out of the column, 8. where the capillaries and porous media is located, 9. the metal holder used to stop the porous media from being ejected from the column due to the pressure build-up beneath the media.

The air injected into the system is kept at a constant flow rate of 50 L/min, this creates foam once it comes in contact with the surfactant solution deposited into the column. The valve was closed once the foam had reached 30 cm up the column. Preliminary testing showed that the foam reached equilibrium and persisted without further change after five minutes. The foam height and liquid height were measured for each system with one camera, while another camera was focused on a prism which is located near the base of the column. This allowed the bubble diameter of the foam to be observed without distortion by the curved column walls.

The foam column can be fitted with different sets of capillaries or with a porous material (for example, the sponges listed in Table 1). The porous materials listed in Table 1 were investigated previously [23] using an SEM device (Loughboroug Materials Characterisation Centre, Loughboroig, United Kingdom). The properties of the capillaries vary in radii size and arrangement. In Figure 2, three sets of capillaries are investigated in two types of arrangements. These are referred below as 'small formation' which is along the top row of Figure 2 and 'large formation' which is positioned along the bottom row.

**Table 1.** Properties of sponge samples found using an SEM device.

| Sponge Type | Pore Size (mm) | Porosity |
| --- | --- | --- |
| Dishwasher | 0.302 ± 0.072 | 0.689 |
| Audio | 0.093 ± 0.028 | 0.692 |
| Car | 0.295 ± 0.070 | 0.694 |

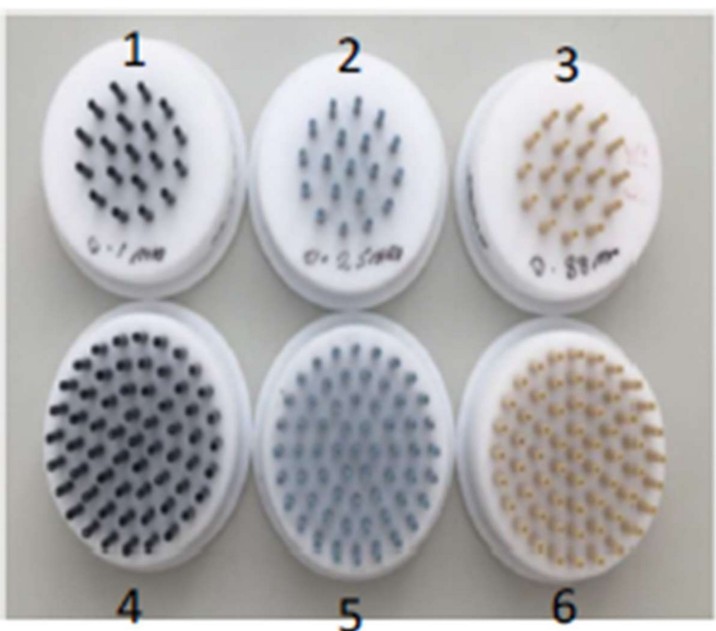

**Figure 2.** The six capillary substrates considered for the project to model incompressible porous substrates of varying pore size and pore arrangements. 1. shows 0.1 mm small arrangement of capillaries, 2. shows 0.25 mm small arrangement of capillaries, 3. shows 0.88 mm small arrangement of capillaries, 4. shows 0.1 mm large arrangement of capillaries, 5. shows 0.25 mm large arrangement of capillaries, and 6. shows 0.88 mm large arrangement of capillaries.

The pore size of the capillaries shown in Figure 2 are as follows: 0.1 mm for the black capillaries, 0.25 mm for the blue capillaries, and 0.88 mm for the cream capillaries.

*2.2. Surfactant Solutions*

SDS and Triton X-100 were both used in the foam column investigations with concentrations of 0.5, 1, 10, 20, and 50 critical micelle concentration (CMC). All the concentrations are clarified in terms of their critical micelle concentration (CMC), where the CMC point for SDS is 8.2 mM and 0.6 mM for Triton X-100 (Sigma-aldrich, Haverhill, United Kingdom). Based on previous investigations, SDS has a similar CMC to the commercial dishwashing product that has been investigated earlier, meaning that 10–50 CMC equates to the concentration range that is predicted for consumer use. The SDS was chosen to be used as our model anionic surfactant due to its wide use in household cleaning products and personal cleaning products. Tween-20 was chosen as our model non-ionic surfactant due to its wide use in the food industry, which involves the production of porous food products which are created by the formation of foams.

SDS was then investigated in more detail using the same concentrations as mentioned previously but now diluted with 15 dH (100 ppm) hard water. Hard water is a mixture of distilled water with salts that would be found in our usual tap water, the measure of hardness is in German degrees (dH), where 1 dH is 0.05603 parts per million (ppm), which allows the investigating of what affect salt content has on foam quality. The salts added and the amounts in grams added are shown in Table 2.

**Table 2.** The salts and amount in grams used to make 2 litres of 15 dH hard water.

| Salt | Mass (g) |
|------|----------|
| $CaCl_{12} \cdot 2H_2O$ | 0.564 |
| $MgCl_2 \cdot 6H_2O$ | 0.300 |
| $NaHCO_3$ | 0.276 |

The SDS with distilled water is also repeated and compared with Tween-20, where a 50/50 mixture at 1 CMC of these surfactants is observed to understand what effect this has on foam quality. The CMC point of Tween-20 is 60 mg/L and is investigated in the same concentrations of 0.5, 1, 10, 20, and 50 CMC.

### 2.3. Average Bubble Diameter and Foam Drainage

The images obtained for bubble diameter and foam height were analysed using ImageJ. For degerming average bubble diameter, it was found that a minimum of 50 bubbles have to been measured to provide the most accurate value of bubble diameter. As indicated by Figure 3, after 50 bubbles, the average bubble diameter remains constant.

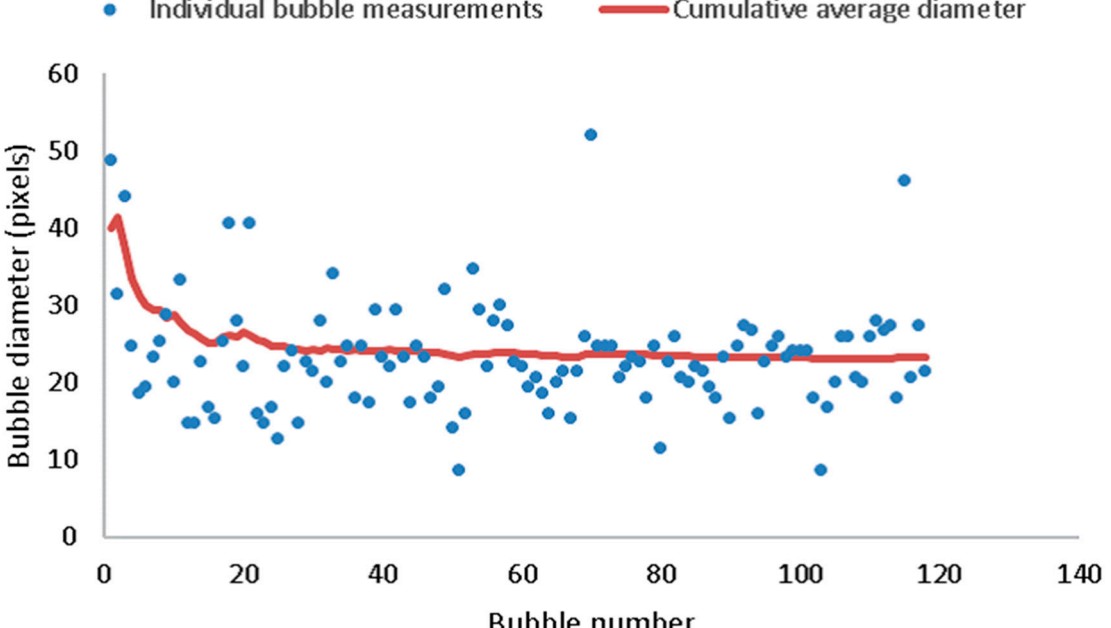

**Figure 3.** The bubble diameter of 118 bubbles along with the cumulative average, indicating that the minimum sample size that can be used is 50 bubbles.

Foam drainage is also investigated using ImageJ, through which the foam height and liquid height are determined in 15 s intervals. The liquid and foam heights are then used to determine the liquid volume fraction of the foam as

$$liquid\ volume\ fraction = \frac{initial\ liquid\ level - liquid\ level\ (t)}{foam\ level\ (t) - liquid\ level\ (t)} \tag{1}$$

Equation (1) is used on the assumption that liquid volume fraction is uniform throughout the foam. However, in reality, this is not the case because it is observed that the top of the foam is much drier than the foam below it, meaning that our values of volume fraction using Equation (1) produced only an averaged liquid volume fraction of the whole foam this led to the use of a homemade dynamic foam analyser in experiments, providing a picture of the liquid volume fraction distribution throughout the foam.

The SDS and Tween-20 solutions are investigated on 0.18 mm capillaries small formation, a dishwasher sponge is then positioned on top of the capillaries. This allows us to

investigate the effects of foam produced with capillaries and its interaction with porous media on foam quality. The capillaries were then removed and the dishwasher sponge is now the only generating media, meaning that, for the second half of the investigation, three separate foaming environments are investigated for SDS and Tween-20. The bubble diameter, foam height, and liquid level are measured for these three systems; this time the liquid volume fraction is investigated using the foam analyser, and then the liquid volume fraction is calculated using Equation (2) (below).

### 2.4. Dynamic Foam Analyser

The resistance of the foam is recorded at four different heights on the foam analyser (made in house at Loughborough University)) using four ohmmeters, taking resistance measurements close to the bottom of the column but still within the foam designated as 0 cm. Three heights are measured within the foam these are 10 cm, 17.5 cm, and 32.5 cm, and the resistance is measured with time as the foam drains. The resistance is then converted to conductivity by doing the inverse of the resistance values. The homemade dynamic foam analyser is shown in Figure 4.

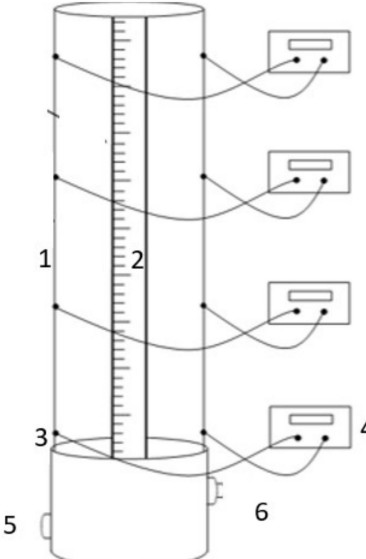

**Figure 4.** Schematic diagram of the homemade dynamic foam analyser, where the first ohm-meter is positioned close to the bottom and the other electrodes are positioned at 10 cm, 17.5 cm, and 32.5 cm above this point. 1. is the acrylic column, 2. scale, 3. electrodes which are located on each side of the column to allow the resistance and hence the conductivity across the foam to measured, 4. ohmmeter used to obtain the resistance across the foam, 5. gas inlet, and 6. liquid outlet which is plugged during the experiment and is used to allow the easy drainage of the liquid out of the column.

The foam is produced by injecting air at a flow rate of 50 L/min until the foam height reaches approximately 34 cm. The resistance at each point is then measured at 1 min, 5 min, and 10 min after generation.

## 3. Results and Discussions

### 3.1. Average Bubble Diameter

The bubble diameter of the foams produced on capillaries and the car sponge (properties which are shown in Table 1) with both SDS and TritonX-100 are shown in Figure 5.

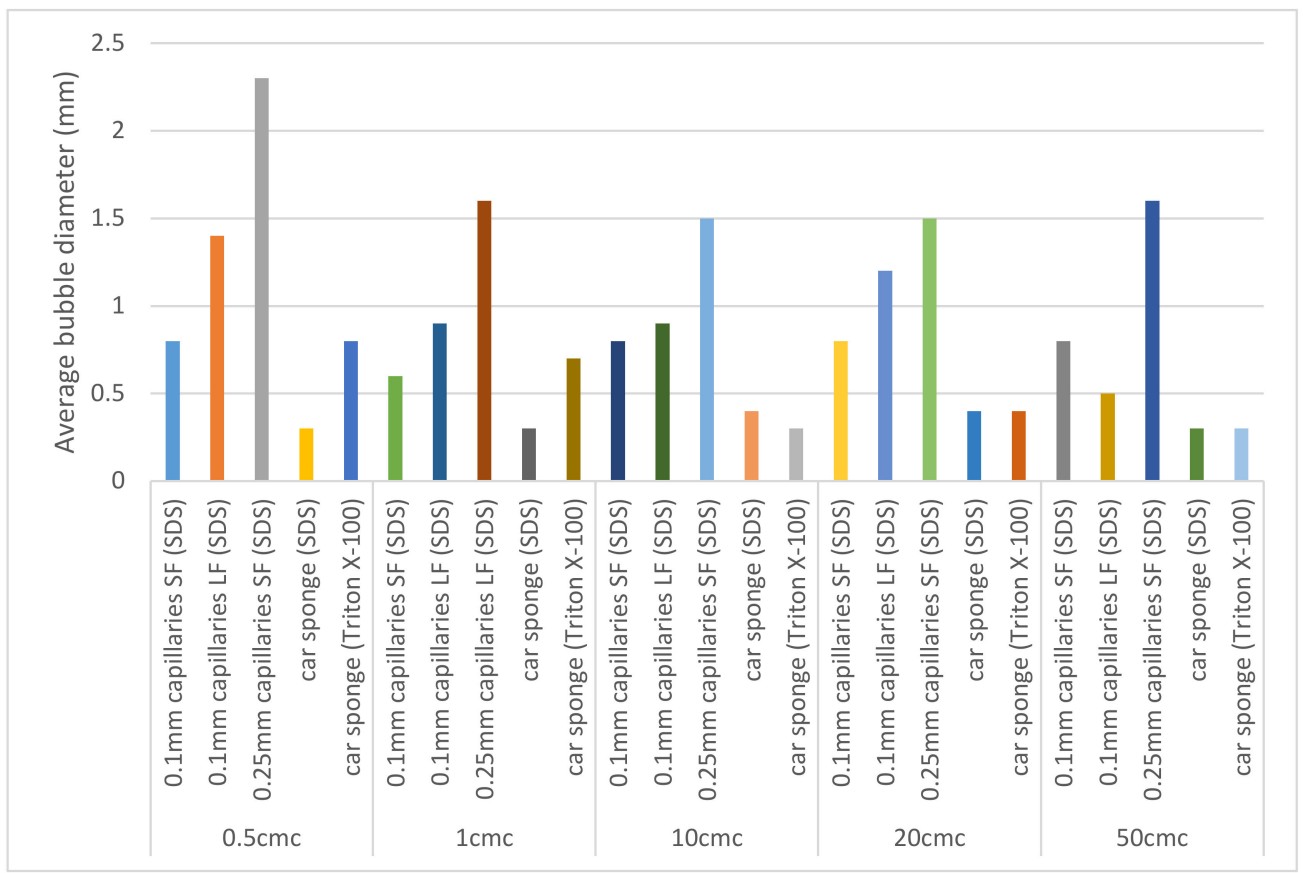

**Figure 5.** The average bubble diameters for each substrate and surfactant type for concentrations 0.5–50 CMC, where SF stands for small formation and LF stands for large formation.

As expected, Figure 5 shows that the highest bubble diameter occurs at 0.5 CMC and that larger capillaries will produce larger bubbles on average. For 0.1 mm capillaries, the small formation configuration showed more homogeneous foams with less fluctuation in average bubble diameter. In contrast, large formation configuration showed heterogeneous foams and significant fluctuation in average bubble diameter values. This is due to the instant coalescence of bubbles due to the close proximity of capillaries to each other leading to a larger average bubble diameter but also a wide variation in bubble diameters present.

The foam produced using the car sponge showed the smallest average bubble diameter, which is surprising due to the trend observed with capillaries that the larger the pore size the larger the average bubble diameter. This could be explained by the 3-D network of pores that exist within the car sponge—there is the issue related to the heterogeneity of the porous network because there is a wide range of pore sizes on the car sponge, meaning there could be a large number of small pores with some larger pores off-setting the average pore size.

### 3.1.1. Dynamic Foam Analyser

The liquid volume fraction is investigated using a homemade dynamic foam analyser. As shown in Figure 6, the advantage of this device is that it allows the liquid volume fraction to be investigated at different heights of the foam and variation over time.

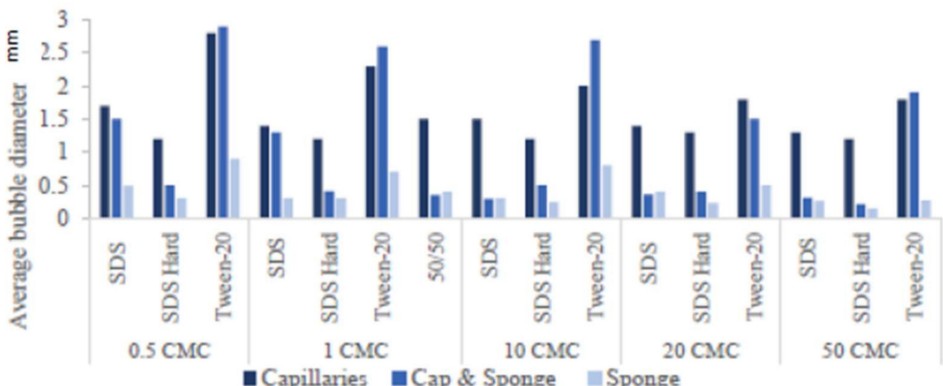

**Figure 6.** Comparison of average bubble diameter under different conditions at concentrations: 0.5, 1, 10, 20, and 50 critical micelle concentration (CMC).

### 3.1.2. Average Bubble Diameter

The average bubble diameter for each surfactant produced on capillaries, capillaries with a dishwasher sponge, and with foam produced by a dishwasher sponge. These results are shown in Figure 6.

Figure 6 shows that introduction of a soft porous media into the capillary system or as the foaming substrate decreases the average bubble diameter. For instance, the average bubble diameter diluted with hard water at a concentration of 50 CMC has a bubble diameter of 1.2 mm with capillaries. In contrast, with a sponge in the capillary system, this drops to 0.21 mm, and when forming foam with the dishwasher sponge instead of capillaries, this drops to 0.15 mm. The bubbles were more homogenous when using capillaries but became more heterogeneous with a wide range of bubble sizes when using a sponge, as shown in Figure 7.

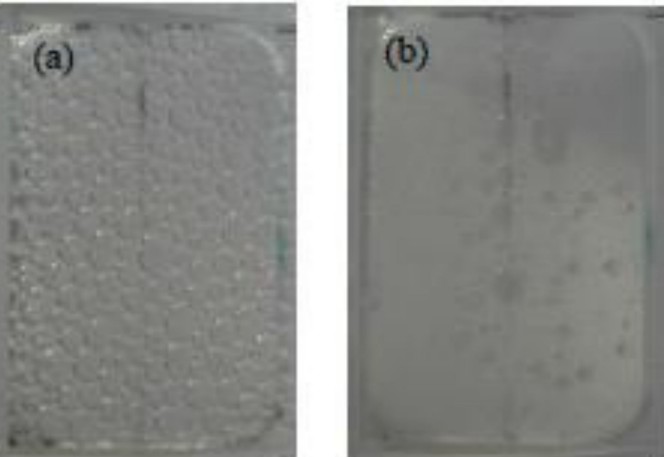

**Figure 7.** Foam produced by sodium dodecyl sulphate (SDS) solutions diluted by hard water at 20 CMC using (**a**) capillaries and (**b**) dishwasher sponge.

In Figure 6, the average bubble diameter for the non-ionic Tween-20 is much larger than the bubble diameter for the anionic SDS. In addition, Tween-20 foam was also more heterogeneous than SDS foam, and the bubbles were polyhedral instead of the usual circular bubbles, indicating that the foam was much drier for Tween-20 than SDS foam.

### 3.1.3. Foam Drainage

The change in foam height with time is shown in Figure 8, where the height of foam decreases with time as indicated by the change in height is shown as negative. This behaviour was expected due to the foam drainage. Initially, as the foam was formed, the

liquid level was not present; once the foam began to drain, the liquid started to collect below the foam. As the result, the foam became more transparent with time, indicating how dry the foam was becoming. Table 3 shows the approximate times taken for the initial liquid level to be increased for each experimental condition. In every instance, the initial liquid level was reached before all the foam had drained, suggesting that the remaining foam is almost completely dry. In Table 3, it can be seen that for the non-ionic Tween-20, the liquid had drained within seconds, whereas for SDS, it took several minutes for all the liquid to drain, indicating as expected that SDS foam is more stable.

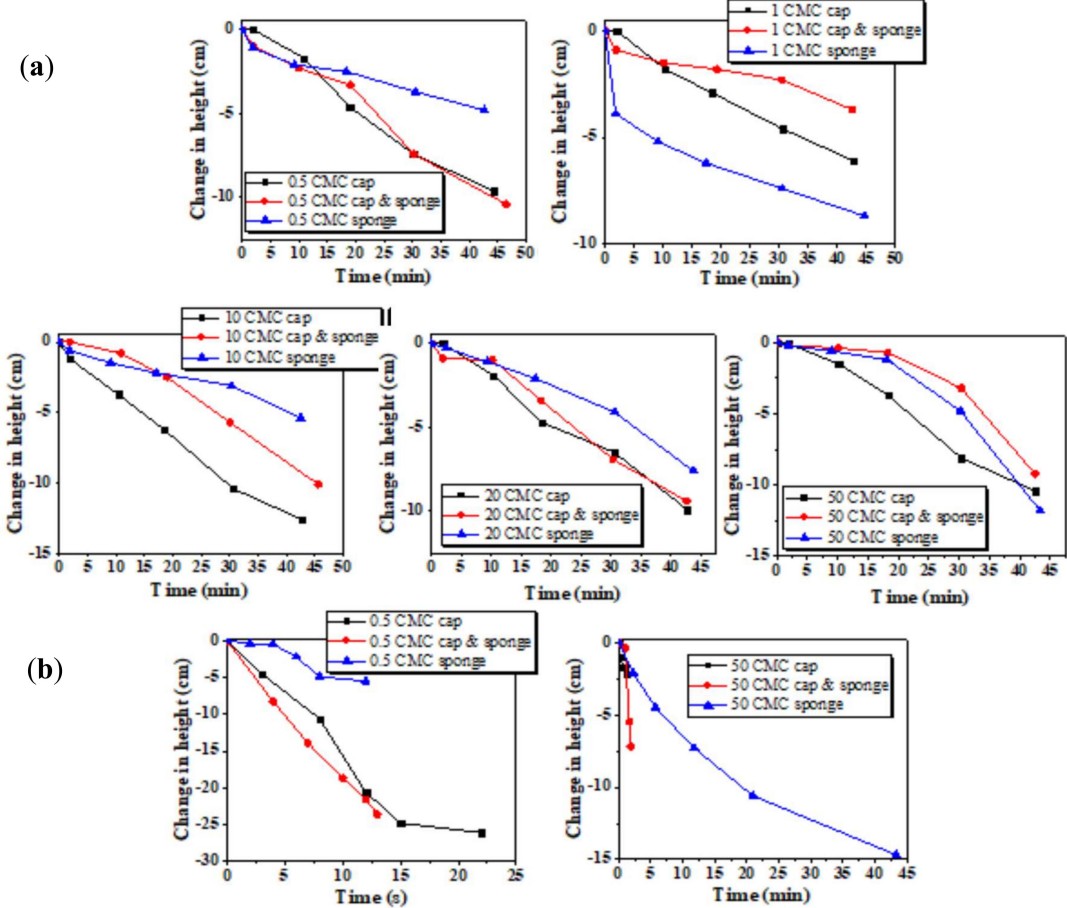

**Figure 8.** Rate of foam drainage on capillaries (cap), capillaries, and dishwasher sponge system and foam formed by dishwasher sponge. (**a**) SDS solutions with hard water at all five concentrations and (**b**) Tween-20 solutions at 0.5 and 50 CMC.

**Table 3.** Time surpassed for the initial liquid level to be reached for each experimental parameter.

| | Time to Reach Final Liquid Level (min) | | | | | | | | |
|---|---|---|---|---|---|---|---|---|---|
| | **SDS (Distilled Water)** | | | **SDS (Hard Water)** | | | **Tween-20** | | |
| **Conc (CMC)** | **Capillaries** | **Cap & Sponge** | **Sponge** | **Capillaries** | **Cap & Sponge** | **Sponge** | **Capillaries** | **Cap & Sponge** | **Sponge** |
| 0.5 | 6 | 0.3 | 6 | 5 | 10 | 8 | 0.3 | 0.1 | 0.1 |
| 1 | 8 | 2 | 14 | 7 | 11 | 15 | 0.1 | 0.1 | 0.3 |
| 10 | 3 | 17 | 10 | 9 | 13 | 26 | 0.5 | 0.3 | 0.7 |
| 20 | 7 | 17 | 14 | 9 | 14 | 28 | 2 | 0.3 | 8 |
| 50 | 5 | 17 | 10 | 10 | 20 | 30 | 2 | 0.3 | 8 |

Figure 8 shows trends in foam drainage for foam produced by capillaries and foam produced by a dishwasher sponge. In Figure 8b, it can be seen that the foam produced with Tween-20 was considerably less stable than that produced by SDS especially at lower concentrations. This is illustrated by the fact that for 0.5 CMC of Tween-20 on sponges, the foam produced would not reach a height above 10 cm, meaning that the low small decrease of height observed in Figure 8b was still a full decay of the foam. An issue observed with the foam decay was that on some occasions coalescence of the bubbles caused the foam to split as shown in Figure 9.

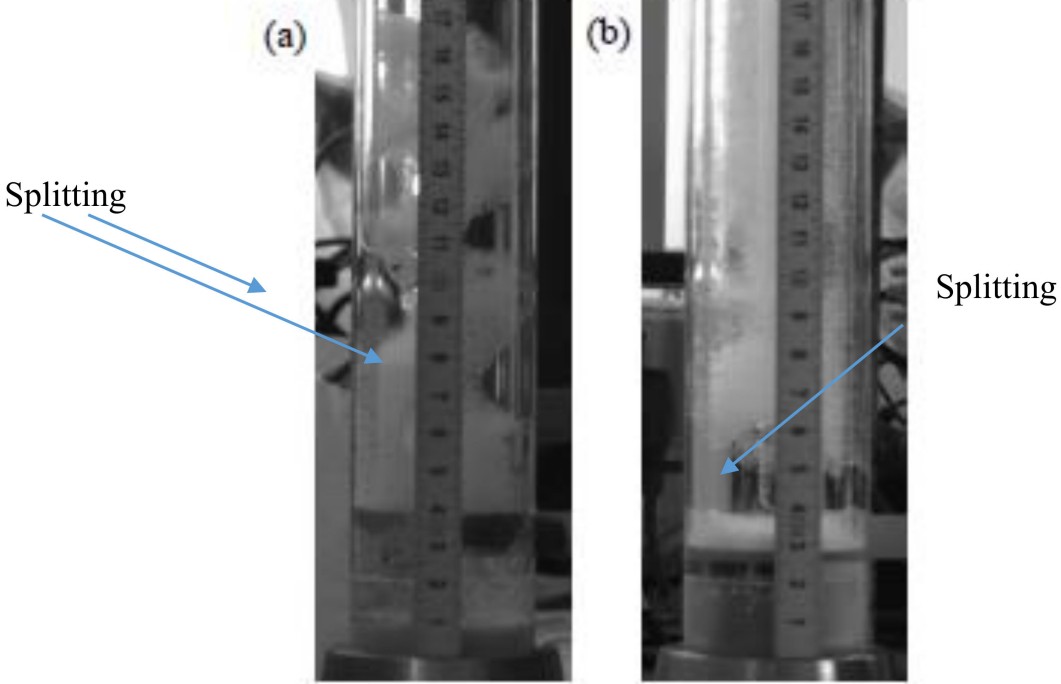

**Figure 9.** Examples of splitting caused by coalescing at (**a**) 1 CMC SDS diluted with distilled water with capillaries and sponge and (**b**) 10 CMC SDS diluted with distilled water with capillaries. The splitting is indicated by the arrow showing the points where the foam has separated.

This splitting was observed for both Tween-20 and SDS with distilled water solutions but was not seen for SDS with hard water solutions.

### 3.2. Liquid Volume Fraction

The conductivity of SDS foams diluted with distilled and 15 dH hard water was investigated using the foam analyser fitted with capillaries, as shown in Figure 4. The conductivity was measured at multiple heights with time and is used in Equation (2) to calculate the liquid volume fraction at each foam height with time, as shown by Figures 10 and 11. According to [30],

$$\sigma = \frac{1}{3}\left(\varphi + \varphi^{\frac{3}{2}} + \varphi^2\right) \qquad (2)$$

where $\sigma$ is the relative conductivity which is the conductivity of the foam normalised by the conductivity of the solution that produced the foam and $\varphi$ is the liquid volume fraction. As indicated by Equation (2), there are three possible solutions for the latter equation. The liquid volume fraction can only be between 0 and 1, meaning that only one possible value of Equation (2) should be selected for each conductivity using this condition. Figures 10 and 11 show the liquid volume fraction against time for SDS solutions diluted by distilled water and 15 dH hard water respectively for each of the heights investigated. It is observed from Figure 11 that the liquid volume fraction of hard water solutions is similar to that of distilled solutions, indicating that this method corrects the effect of the difference

in ion concentration when comparing different foaming solutions. Figures 10 and 11 show as expected that the higher up the foam, the drier it becomes and that the liquid volume fraction is not uniform. Figures 10a–d and 11a–d all follow the fitted relationship (Equation (3)), whereas, for the 50 CMC results, the first five min did not drain exponentially and had a surprisingly high liquid volume fraction value of 0.7, indicating that this point is located inside an emulsion, not a foam. After 10 min, the liquid volume fraction has dropped to around 0.35 and follows the exponential decay relationship (Equation (3)), indicating that the emulsion of SDS solution with air has decayed into a foam. Figure 12 shows that for both distilled and 15 dH hard water solutions, as the concentration of SDS increases, the liquid volume increases in turn. Although, as shown by 50 CMC, this eventually leads to the production of an emulsion with liquid volume fraction values as high as 0.7 which leads to a longer time for drainage which may not be desired for consumer applications particularly with cleaning because it will take too long for the product to deposited on the area being treated.

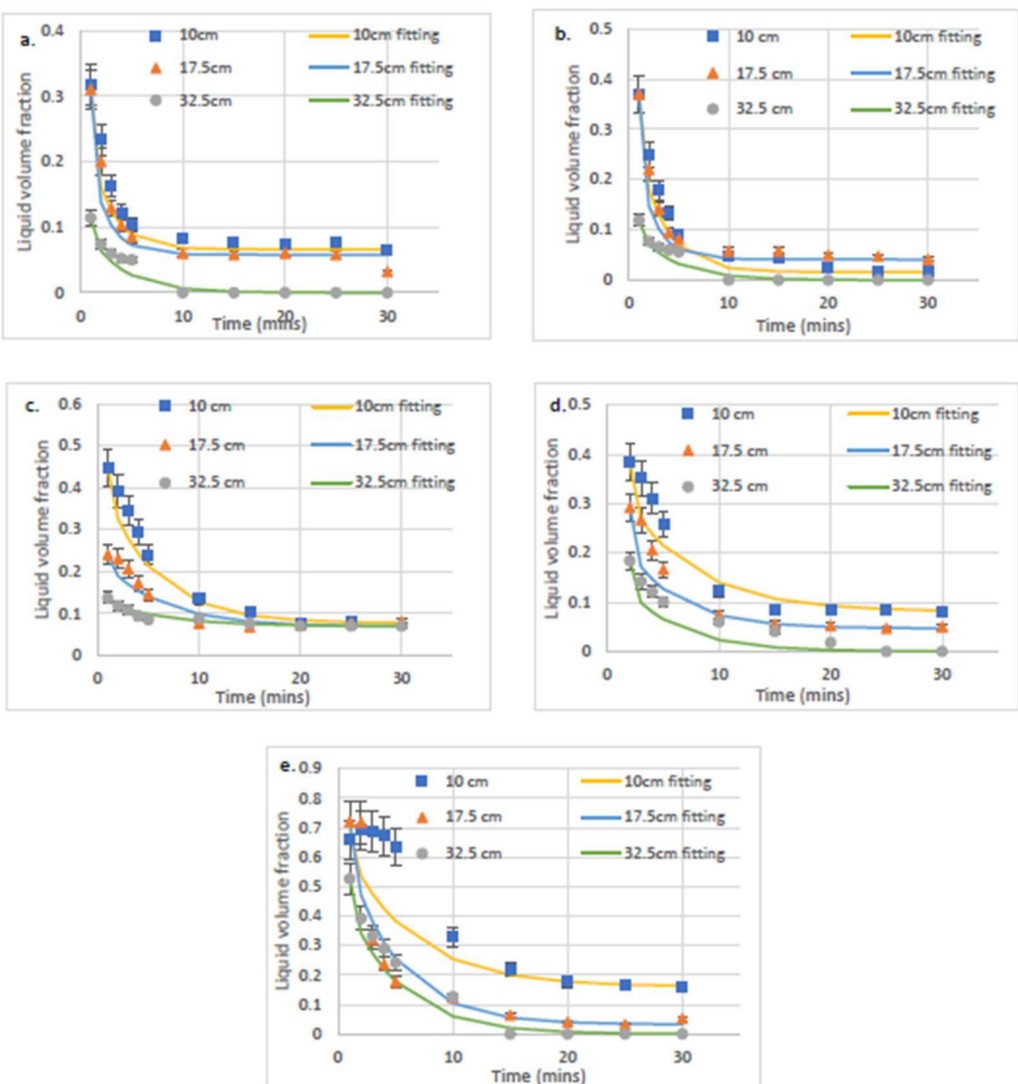

**Figure 10.** Dependency of liquid volume fraction with time of foam at each height for SDS solutions diluted with distilled water with concentrations (**a**) 0.5 CMC, (**b**) 1 CMC, (**c**) 10 CMC, (**d**) 20 CMC, and (**e**) 50 CMC. The fitting is according to Equation (3).

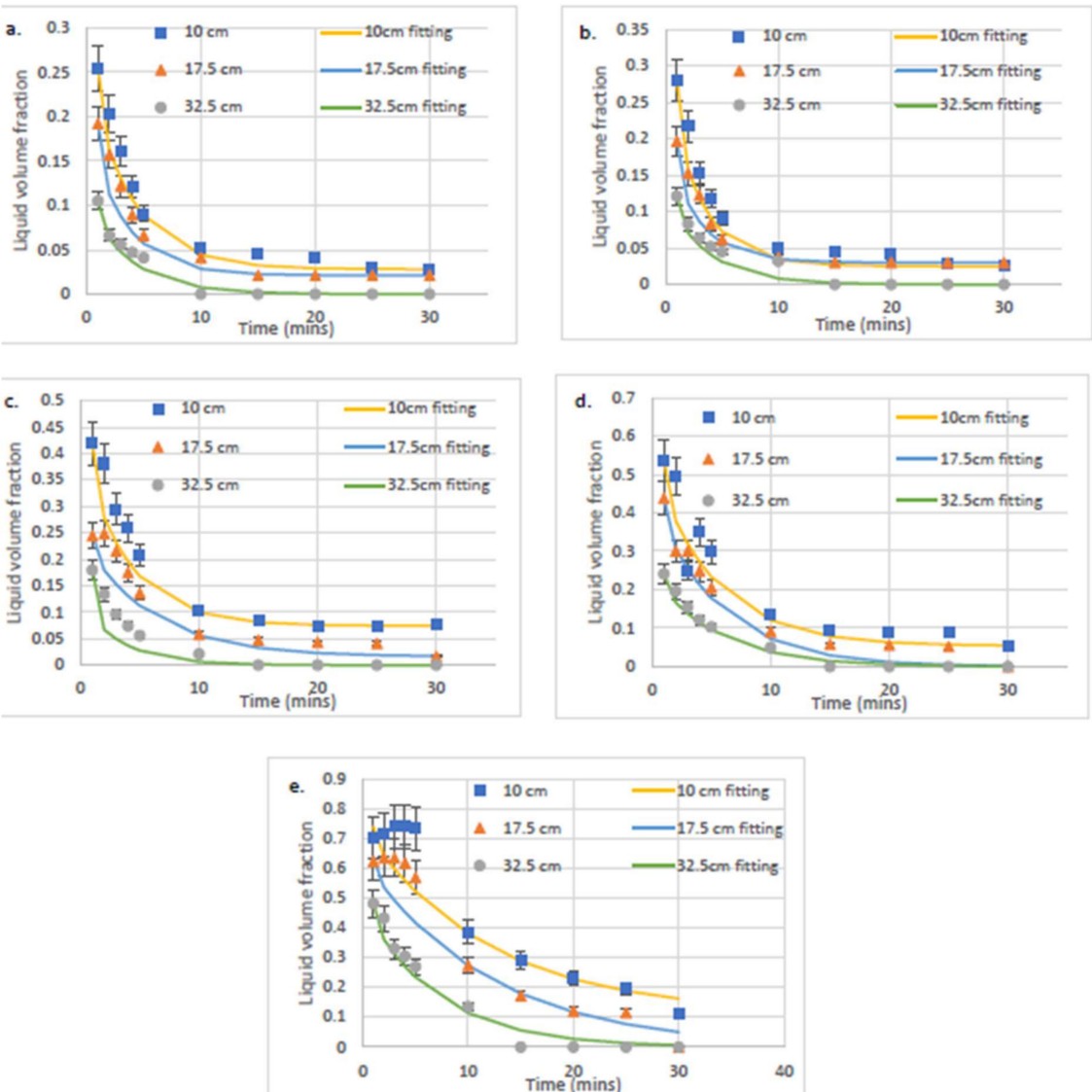

**Figure 11.** Dependency of liquid volume fraction with time of foam at each height for SDS solutions diluted with distilled 15dH hard water with concentrations (**a**) 0.5 CMC, (**b**) 1 CMC, (**c**) 10 CMC, (**d**) 20 CMC, and (**e**) 50 CMC. The fitting is according to Equation (3).

The experimental data displayed in Figures 10–12 are fitted using Equation (3) as follows:

$$\varphi(t) \; = \; \varphi_\infty + \; (\varphi_0 \; - \; \varphi_\infty)e^{(-\alpha t)} \tag{3}$$

where $\varphi(t)$ is the liquid volume fraction at the moment t, $\varphi_0$ and $\varphi_\infty$ are initial and final liquid volume fraction; $1/\alpha$ is a characteristic time scale of the process. Equation (3) can be rewritten as

$$\ln \left\{ [\varphi(t) - \varphi_\infty]/(\varphi_0 - \varphi_\infty) \right\} = \; -\alpha t \tag{4}$$

from the slope of the latter dependency, the time scale $1/\alpha$ can be extracted. The experimental results showed good agreement with the derived mathematical fitting as demonstrated by Figures 10–12.

The extracted characteristic time scales according to Equation (4) are shown in Tables 4 and 5. Table 4 shows the characteristic time for each height for SDS solutions with distilled water and Table 5 shows the characteristic time for SDS solutions with 15 dH hard water.

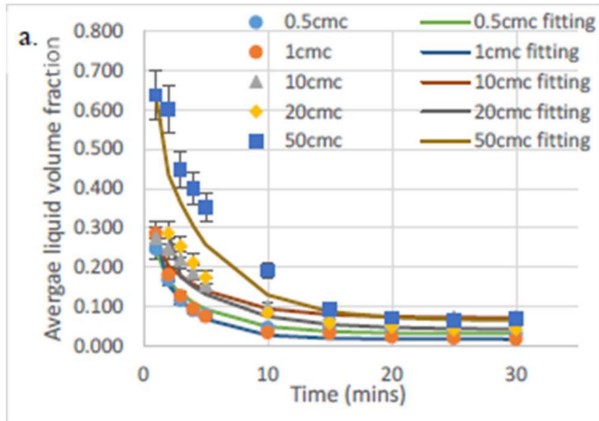
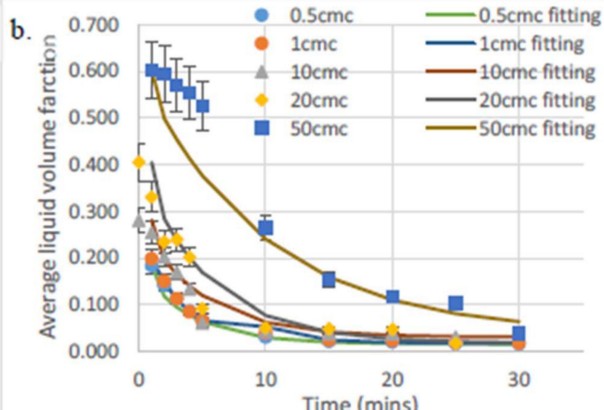

**Figure 12.** Average liquid volume fraction for whole foam against time for each concentration of SDS, where (**a**) is distilled water solutions and (**b**) is 15 dH hard water solutions. The fitting is according to Equation (3).

**Table 4.** The characteristic time scale obtained using Equation (4) for foams produced with SDS solutions with distilled water for each of the heights investigated.

| Concentration (CMC) | Height (cm) | Alpha (1/min) | 1/Alpha (min) | 1/Alpha (s) |
|---|---|---|---|---|
| 50 | 10 | 0.1736 | 5.7604 | 346 |
|  | 17.5 | 0.2175 | 4.5977 | 276 |
|  | 32.5 | 0.2255 | 4.4346 | 266 |
| 20 | 10 | 0.164 | 6.0976 | 366 |
|  | 17.5 | 0.2242 | 4.4603 | 268 |
|  | 32.5 | 0.2069 | 4.8333 | 290 |
| 10 | 10 | 0.2 | 5.0000 | 300 |
|  | 17.5 | 0.1741 | 5.7438 | 345 |
|  | 32.5 | 0.3505 | 2.8531 | 171 |
| 1 | 10 | 0.3811 | 2.6240 | 157 |
|  | 17.5 | 0.5632 | 1.7756 | 107 |
|  | 32.5 | 0.262 | 3.8168 | 229 |
| 0.5 | 10 | 0.5737 | 1.7431 | 105 |
|  | 17.5 | 0.4816 | 2.0764 | 125 |
|  | 32.5 | 0.2908 | 3.4388 | 206 |

It can be derived from Tables 4 and 5 that, other than 50 CMC, the characteristic time for all the concentrations for both distilled water and 15 dH hard water are reasonably similar. The large difference observed with 50 CMC is due to initially the liquid volume fraction of the foam is so large that initially, indicating that for 50 CMC this is an emulsion until drainage eventually reaches around 0.36 liquid volume fraction when it starts to behave as a foam and follows the trend described by Equation (3). This behaviour was observed more clearly in the case of 15 dH hard water where the higher stability means the emulsion stage remains longer and affects the characteristic time dramatically.

The good fitting according to Equation (4) gave us the idea to try the same dependency for the averaged volume fraction according to Equation (2). The result is shown in Figure 12, where the characteristic time scales are shown in Tables 6 and 7. Table 6 shows SDS solutions with distilled water, and Table 7 shows SDS solutions with 15 dH hard water.

**Table 5.** The characteristic time scale obtained using Equation (4) for foams produced with SDS solutions with 15 dH hard water for each of the heights investigated.

| Concentration (CMC) | Height (cm) | Alpha (1/min) | 1/Alpha (min) | 1/Alpha (s) |
|---|---|---|---|---|
| | 10 | 0.0842 | 11.8765 | 713 |
| 50 | 17.5 | 0.0854 | 11.7096 | 703 |
| | 32.5 | 0.1436 | 6.9638 | 418 |
| | 10 | 0.1864 | 5.3648 | 322 |
| 20 | 17.5 | 0.1804 | 5.5432 | 333 |
| | 32.5 | 0.1864 | 5.3648 | 322 |
| | 10 | 0.2605 | 3.8388 | 230 |
| 10 | 17.5 | 0.1759 | 5.6850 | 341 |
| | 32.5 | 0.2963 | 3.3750 | 202 |
| | 10 | 0.3302 | 3.0285 | 182 |
| 1 | 17.5 | 0.3603 | 2.7755 | 167 |
| | 32.5 | 0.2714 | 3.6846 | 221 |
| | 10 | 0.2634 | 3.7965 | 228 |
| 0.5 | 17.5 | 0.313 | 3.1949 | 192 |
| | 32.5 | 0.2632 | 3.7994 | 228 |

**Table 6.** The characteristic time scale obtained using Equation (4) for foams produced with SDS solutions with distilled water.

| Concentration (CMC) | Alpha (1/min) | 1/Alpha (min) | 1/Alpha (s) |
|---|---|---|---|
| 50 | 0.2175 | 4.5977 | 276 |
| 20 | 0.2024 | 4.9407 | 296 |
| 10 | 0.2165 | 4.6189 | 277 |
| 1 | 0.3329 | 3.0039 | 180 |
| 0.5 | 0.25 | 4.0000 | 240 |

**Table 7.** The characteristic time scale obtained using Equation (4) for foams produced using SDS solutions with 15 dH hard water.

| Concentration (CMC) | Alpha (1/min) | 1/Alpha (min) | 1/Alpha (s) |
|---|---|---|---|
| 50 | 0.1027 | 9.7371 | 584 |
| 20 | 0.1879 | 5.3220 | 319 |
| 10 | 0.2069 | 4.8333 | 290 |
| 1 | 0.3302 | 3.0285 | 182 |
| 0.5 | 0.2516 | 3.9746 | 238 |

It can be derived from Tables 6 and 7 that, other than 50 CMC, the characteristic time for all the concentrations for both distilled water and 15 dH hard water are reasonably similar, as discussed previously in Tables 4 and 5.

## 4. Conclusions

The average bubble diameter of foam was larger for 0.25 mm capillaries compared to 0.1 mm capillaries and capillary configuration affected the homogeneity of the foam and the range of bubble diameters that were present. The car sponge produced the foam with the smallest average bubble diameter, which is due to the heterogeneity of the porous media and the complex 3-D network of the substrate. It was found that foam formed with soft porous media produces foam with a smaller average bubble diameter compared to capillaries when using both anionic and non-ionic surfactant solutions. Foam formed on just capillaries is very homogenous with a small range of bubble sizes and have similar shapes. When formed through a sponge, the shape remains consistent, but there is a larger range of bubble sizes produced.

There is no set trend between the decay of foam produced in capillaries versus soft porous media, indicating that differences in the decay of foam in this investigation are more dependent on the environment such as temperature variations. Foam decays at an increased rate at lower concentrations of Tween-20 compared to SDS (produced with both distilled and hard water), with a full decay taking seconds compared to hours. Foam decay normally occurred from the top of the foam downwards, however, in some instances splitting of the foam happened due to the coalescence of the bubbles.

A relationship between liquid volume fraction and conductivity has been demonstrated and explored using the foam analyser. As expected, it was found that the higher up the column the lower the conductivity and hence the drier the foam. There was no observed difference between the liquid volume fraction values of distilled water SDS solutions and hard water SDS solutions. A fitting relationship was derived for the decay of liquid volume fraction with time which showed good agreement with concentrations of SDS 0.5 CMC to 20 CMC for both distilled and 15 dH hard water solutions, whereas for the 50 CMC results the first five min did not drain exponentially and had a surprisingly high liquid volume fraction value of 0.7, indicating at this point that the substance is an emulsion, not a foam. After 10 min, the liquid volume fraction of 50 CMC has dropped to around 0.35 and follows the exponential decay relationship observed previously discussed for 0.5–20 CMC. For both distilled and 15 dH hard water solutions, as the concentration of SDS increases, the liquid volume increases in turn, hence indicating an increase in the overall quality of the foam. Although, as shown by 50 CMC, there is a limit to when the foam turns into emulsion caused by this increase restriction to drainage, creating a product that may be undesirable for application, which decreases its usefulness as a delivery mechanism due to the decreased drainage rate.

**Author Contributions:** Conceptualisation, V.S. and A.T.; methodology, A.T. and P.J.; validation, A.T. and P.J.; formal analysis, P.J.; investigation, P.J.; resources, A.T. and M.V.; data curation, P.J.; writing—original draft preparation, P.J.; writing—review and editing, A.T., V.S., M.V. and P.J.; visualisation, P.J.; supervision, A.T.; project administration, A.T. and M.V.; funding acquisition, A.T. and V.S. All authors have read and agreed to the published version of the manuscript.

**Funding:** This research was supported by Proctor & Gamble, Brussels, and Map Evaporation Project, ESA and NanoPaint, Marie Curie EU project.

**Acknowledgments:** We would also acknowledge the hard work conducted by Toni Alana Robertson, Louise Tomkins, who were students at Loughborough University and provided useful input in the production of this manuscript. The authors acknowledge use of the facilities within the Loughborough Materials Characterisation Centre.

**Conflicts of Interest:** The authors declare no conflict of interest.

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
