# Peer review of "Foam Quality of Foams Formed on Capillaries and Porous Media Systems"

_colloids, doi:10.3390/colloids5010010_

Round 1

Reviewer 1 Report

The article is about the foam creation in capillary and porous media. The first part of the study is about foam creation by capillary in a tube, bubble size is characterised. These are standard measurements done to characterise foam. the conclusions are not bringing new insights. 

Consequently , it is not recommended for publication.

Some remarks are listed below:

Abstract: It is overstated that "Interaction of foams with porous media has only recently been investigated by O. Arjmandi-Tash et al."  Foam in porous media is a reach field of research for 50 years, and O. Arjmandi-Tash et al. only did a modest investigation. Authors should reword this sentence because it is misleading. 

Page 4, air is injected at fixed pressure , however it seems in figure 2 that the permeability of the substrates are different, meaning that the volume of gas injected is not the same in each substracts. Authors should than explain if flow rate of gas injected is constant for all experiments or , if not, how to compare experiments.

Author Response

Thank you for your comments and suggestions, changes made in the manuscript are marked by track changes and response each of your comments are found below:

The article is about the foam creation in capillary and porous media. The first part of the study is about foam creation by capillary in a tube, bubble size is characterised. These are standard measurements done to characterise foam. the conclusions are not bringing new insights. 

Consequently, it is not recommended for publication.

In this study we have investigated the affects capillary arrangement and how different arrangements but with the same capillary size affects the average bubble diameter of the produced foam. These results are then compared to the foam produced using a porous media with the objective to find a capillary size and arrangement that would be similar to the foam produced by the porous media.

Even though these measurements are standard in the characterisation of foam in the literature we have encountered there is not a comprehensive investigation of the quality of these foams and this had to be clarified using a well know system, such as capillaries, to fully comprehend the affects a more complex system has on the quality of foam. It is then important that the first part of the investigation is included as this facilitates the results required to compare with the porous media. These investigations also indicate what next is required when trying to form a model of the quality of foam produced when using a porous media and highlights the affect the interconnectivity and heterogeneity of the media has on the quality of foam.

Some remarks are listed below:

Abstract: It is overstated that "Interaction of foams with porous media has only recently been investigated by O. Arjmandi-Tash et al."  Foam in porous media is a reach field of research for 50 years, and O. Arjmandi-Tash et al. only did a modest investigation. Authors should reword this sentence because it is misleading. 

This has been reworded as requested

Page 4, air is injected at fixed pressure, however it seems in figure 2 that the permeability of the substrates are different, meaning that the volume of gas injected is not the same in each substracts. Authors should than explain if flow rate of gas injected is constant for all experiments or , if not, how to compare experiments.

Apologies for the miss understanding the flow rate in the experiments of the injected gas was kept at a constant 50 L/min which was mentioned after figure 4 for the dynamic foam analyser. The discussion about pressure was for the initial pressure of the gas released into the system which had to be reduced for the porous media to avoid the porous media escaping the column. Though these details are not important for the investigation and have now been removed and as suggested only the injected flow rate will be discussed which was the same for all the investigations.

Reviewer 2 Report

The submitted manuscript entitled ‘Foam quality of foams formed on capillaries and porous media systems’ is dealing with the foam formation and behaviour (average bubble diameters, stability, drainage, splitting, etc.). Capillaries and soft porous media were used as a gas inlet in the foaming process. The manuscript is interesting, the main concern f this Reviewer is in the application of the results. Could the Authors highlight any given real-life applications, where these results are really important? Besides, the following technical issues arose.

- Keywords are missing, please add 3-10 relevant keywords.

- The Introduction mentions 24 references, However, most of them have been mentioned only in largely general style. Please detail the relevant information from the references.

- Fig 1 is not a diagram, rather drawings or sketches. Please label the subfigs accordingly ((a), (b), etc.). The caption is too long, please move the description to the main text.

- Please always let a space between the value and its unit, except in the case of ‘°C’ and ‘%’.

- Please write in millimetres instead of centimetres.

- Eq 1 should be rewritten into a single line equation.

- Fig 4: there are white rectangles in the drawing (presumably shading texts…), please redraw the figure. If it was published earlier, please add a reference. The same arose here connected to the right part of fig 1. The caption of fig 4 is too long, please move the description to the main text.

- Fig 8 is very low in quality (blurry) and should be redrawn.

- Please indicate the split parts in fig 9.

- Line 253: please start the section title by capital ‘L’.

- Please use exponential forms instead of ‘exp’ in the equations.

- Table 7 is broken, please rearrange the text to solve the problem.

- The Authors’ contributions are missing, please fill in the corresponding paragraph (lines 353-359).

- Please fill in or delete the section ‘Funding’ (lines 360-363).

- Please fill in the section ‘Conflict of Interest’ (lines 368-374).

Author Response

The submitted manuscript entitled ‘Foam quality of foams formed on capillaries and porous media systems’ is dealing with the foam formation and behaviour (average bubble diameters, stability, drainage, splitting, etc.). Capillaries and soft porous media were used as a gas inlet in the foaming process. The manuscript is interesting, the main concern f this Reviewer is in the application of the results.

Thank you for your comments and suggestions, changes made in the manuscript are marked by track changes and response each of your comments are found below:

Could the Authors highlight any given real-life applications, where these results are really important? Besides, the following technical issues arose.

The work conducted here is of great importance in understanding the quality foams produced in porous media the industries which will find use or build upon the data here is the house cleaning industry in particular dishwashing. Also, the understanding of foam formation by air injection through porous media is of great interest for enhanced oil recovery (EOR.) Though the media which will be of interest for EOR will be a lot harder and non-deformable which is an area which could be built upon using this work using similar methods.

These examples have been discussed in the introduction section and as requested later references that relate to this have now been expanded on to further sate this fact.

- Keywords are missing, please add 3-10 relevant keywords.

This has now been included

- The Introduction mentions 24 references, However, most of them have been mentioned only in largely general style. Please detail the relevant information from the references.

Information from some of the references has been expended on in particular the references discussing the interaction and formation of foam with porous media to address the first point as well.

- Fig 1 is not a diagram, rather drawings or sketches. Please label the subfigs accordingly ((a), (b), etc.). The caption is too long, please move the description to the main text.

This has been reworded from diagrams to drawings, the subfigure labels have also been changed to (a) and (b). As suggested part of the caption has been moved to the main text, where the explanation of the labels for figure 1 are now discussed in the main text and not in the caption.

- Please always let a space between the value and its unit, except in the case of ‘°C’ and ‘%’.

This has been edited as suggested.

-Please write in millimetres instead of centimetres.

In the manuscript we thought it best to discuss the foam height in terms of cms and the bubble diameter in terms mm. Due to the figures and graphs being in terms of cm for foam height we wish to keep the units in cm when discussing foam height.

- Eq 1 should be rewritten into a single line equation.

This has been corrected

- Fig 4: there are white rectangles in the drawing (presumably shading texts…), please redraw the figure. If it was published earlier, please add a reference. The same arose here connected to the right part of fig 1. The caption of fig 4 is too long, please move the description to the main text.

Figure 4 has now been redrawn and the white rectangles have been removed, the text concerning what the number represent in the figure has been moved to the main text to reduce the size of the caption as requested.

- Fig 8 is very low in quality (blurry) and should be redrawn.

Figure 8 has been redrawn as requested and quality of the image has been improved.

- Please indicate the split parts in fig 9.

Arrows have been added to the figure to indicate the points where the splitting has occurred with additional text in the caption to indicate that the arrows demonstrate the points the splitting has occurred.

- Line 253: please start the section title by capital ‘L’.

This has been corrected

- Please use exponential forms instead of ‘exp’ in the equations.

This has been changed as requested.

- Table 7 is broken, please rearrange the text to solve the problem.

This has been corrected

- The Authors’ contributions are missing, please fill in the corresponding paragraph (lines 353-359).

This has now been added.

- Please fill in or delete the section ‘Funding’ (lines 360-363).

This has now been added

- Please fill in the section ‘Conflict of Interest’ (lines 368-374).

This has now been added

Reviewer 3 Report

The manuscript presents experimental data about the effect of capillary sizes and the specific conditions of generating the foam at the surface of the porous media on the quality of foams. Two model cases are investigated: aqueous solutions of Tween-20 and SDS.

The manuscript is of interest to the audience of MDPI Colloids and interfaces. Before publication however, the authors should consider the following issues:

  1. A list of keywords has to be supplied.
  2. The particular choice of the model surfactants should be explained and substantiated.
  3. On p.10, line 257 the statement “According to (30)..” is unclear. Is it Ref. [30]? If yes, there is no such a Reference number in the list.

Author Response

The manuscript presents experimental data about the effect of capillary sizes and the specific conditions of generating the foam at the surface of the porous media on the quality of foams. Two model cases are investigated: aqueous solutions of Tween-20 and SDS.

The manuscript is of interest to the audience of MDPI Colloids and interfaces. Before publication however, the authors should consider the following issues:

Dear whomever it may concern,

Thank you for your comments and suggestions, changes made in the manuscript are marked by track changes and response each of your comments are found below:

1. A list of keywords has to be supplied.

This has now been included

2. The particular choice of the model surfactants should be explained and substantiated.

The surfactants where chosen so that during the experiment we had one anionic surfactant and one non-ionic surfactant. Also, SDS was chosen as it is one of the most widely used surfactants in house and personal cleaning products. Tween-20 was used as this surfactant is widely used within the food industry and since some products within food industry involves the formation of porous products by the formation of foams. This provides a different perspective at which the data from these investigations could be made useful. A brief explanation of this has been added to the materials and methods section.

3. On p.10, line 257 the statement “According to (30)..” is unclear. Is it Ref. [30]? If yes, there is no such a Reference number in the list.

Apologies and thank you for spotting this there should be a reference 30 and this has now been added into the text, where ref 30 is:  Weaire D, Hutzler S. The Physics of Foams. Oxford University press; 1999.

Round 2

Reviewer 1 Report

The reviewers comments have been accounted in the manuscript. As the authors relate the study with EOR , it would be important to place this study in this context.

The effect of the surfactant concertation on the foaming and the foam flow in porous media have been studied by Jones et al [Journal of Industrial and Engineering Chemistry Volume 37, 25 May 2016, Pages 268-276] showing that, for the foam to be stable in porous media, the surfactant concentration needs to be at much higher concentration (x100) than the CMC. How is it related to the outcomes of this study? 

Author Response

Response to reviewer 1

The reviewers comments have been accounted in the manuscript. As the authors relate the study with EOR , it would be important to place this study in this context.

The effect of the surfactant concertation on the foaming and the foam flow in porous media have been studied by Jones et al [Journal of Industrial and Engineering Chemistry Volume 37, 25 May 2016, Pages 268-276] showing that, for the foam to be stable in porous media, the surfactant concentration needs to be at much higher concentration (x100) than the CMC. How is it related to the outcomes of this study?

Dear whomever it may concern thank you for your comment.

To answer your question the way this works relates to EOR is by the basic process of foam injection into a porous media and many of the interactions that are observed in EOR will occur in this experimental setup. The difference in stability is due to the media that is being used in traditional EOR these are porous rocks which will have a much smaller pore size and will have a much lower porosity than the media investigated here. In addition, the porous rocks internal structure will be completely different and the difference in interconnectivity and dead-end pores will lead to larger amount of surfactant needed to produce a stable foam. Finally, the foam stability was observed by us to heavily dependent on the porous media used meaning due to the different medias involved in each investigation this will explain the difference in concentration required to produced stable foam.

Reviewer 2 Report

Thank you for all the corrections and modifications.

In the opinion of this Reviewer, the manuscript is ready for publication; however, the final decision belongs to the academic editor of course.

Author Response

Thank you for all the corrections and modifications.

In the opinion of this Reviewer, the manuscript is ready for publication; however, the final decision belongs to the academic editor of course.

Thank you.